# CLEVRER-Humans: Describing Physical and Causal Events the Human Way

**Jiayuan Mao**[*]
MIT

**Xuelin Yang**[*]
Stanford University

**Xikun Zhang**
Stanford University

**Noah D. Goodman**
Stanford University

**Jiajun Wu**
Stanford University

## Abstract

Building machines that can reason about physical events and their causal relationships is crucial for flexible interaction with the physical world. However, most existing physical and causal reasoning benchmarks are exclusively based on synthetically generated events and synthetic natural language descriptions of causal relationships. This design brings up two issues. First, there is a lack of diversity in both event types and natural language descriptions; second, causal relationships based on manually-defined heuristics are different from human judgments. To address both shortcomings, we present the CLEVRER-Humans benchmark, a video reasoning dataset for causal judgment of physical events with human labels. We employ two techniques to improve data collection efficiency: first, a novel iterative event cloze task to elicit a new representation of events in videos, which we term Causal Event Graphs (CEGs); second, a data augmentation technique based on neural language generative models. We convert the collected CEGs into questions and answers to be consistent with prior work. Finally, we study a collection of baseline approaches for CLEVRER-Humans question-answering, highlighting the great challenges set forth by our benchmark.

## 1 Introduction

The ability to reason about physical events and their causal relationships from visual observations lies at the core of human intelligence. It is crucial for humans to holistically understand and flexibly interact with the physical world [1, 2, 3, 4, 5]. Natural language provides a way for humans to express such causal understanding [6], and we can use natural language as a lens to evaluate machine understanding of physical events and causal judgments. This brings us two major advantages. First, compared to bounding boxes and timecodes, language provides a more flexible interface for describing events. Furthermore, it naturally enables the generation of human-interpretable explanations.

We have seen significant progress toward machine reasoning about physical events and causal structures, driven by datasets such as CLEVRER [7] and CATER [8]. These datasets pair videos containing physical object interactions (e.g., collisions) with natural language question-answer pairs. However, there are two important design flaws in existing datasets. First, the categories of physical events are designed manually and identified through heuristic rules over object locations and velocities, significantly restricting the datasets' diversity. For example, the CLEVRER dataset only contains three types of events (object enter, object exit, and collision). Second, the causal relationships of

---

[*]indicates equal contribution. Authors ordered alphabetically. Correspondence to: `jiayuanm@mit.edu`.
Project page: https://sites.google.com/stanford.edu/clevrer-humans/home.
Dataset DOI: https://doi.org/10.5061/dryad.5tb2rbp7c.

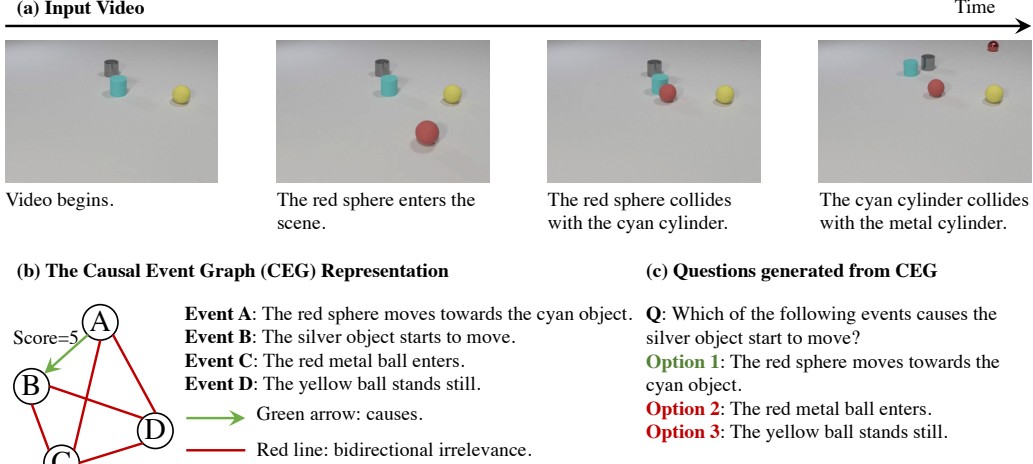

Figure 1: For (a) each video in the CLEVRER dataset, (b) CLEVRER-Humans annotates a human-labeled graphical representation of physical events and their causal relations, in the form of causal event graphs (CEGs). Each CEG composes of a collection of nodes associated with textual descriptions of events, and human-labeled directional edges indicating the causal relationship between objects. Each edge is also associated with a score indicating a human's graded attribute of causal relations. (c) The compact representation of CEGs can be easily translated into question-answer pairs to evaluate video reasoning models.

events are also detected by heuristic rules. In contrast, human causal judgment resembles complex counterfactual reasoning processes [9, 2, 6], and it is difficult to capture these subtleties through manually specified rules.

To address both shortcomings, we present CLEVRER-Humans, a human-annotated physical and causal reasoning dataset based on videos from CLEVRER [7], of which we show examples in Fig. 1a. To ensure sufficient event diversity and annotation density (i.e., a dense causal relation between events), CLEVRER-Humans relies on a data representation termed Causal Event Graphs [CEG; 7, 10], whose nodes are natural language descriptions of the events in the video and edges are causal relationships between the events, as illustrated in Fig. 1b. The CEG annotation procedure has two stages. In the first stage, we use an iterative event cloze task to collect event descriptions. Specifically, starting from a seeding set of events in the original CLEVRER dataset, we ask annotators to describe other events that are responsible for these seed events. The newly annotated events will be iteratively used as new seeds to progressively grow the annotated events. In the second stage, we condense the CEGs by asking human annotators to make binary classifications of causal influence for all pairs of physical events generated in the first stage. Based on both positive and negative labels, we can construct physical and causal reasoning questions in natural language, shown in Fig. 1c.

One challenge of this pipeline is that human annotation for event cloze tasks is generally time- and budget-consuming. To alleviate this, we leverage the observation that neural generative models are reasonably good at generating event descriptions. Thus, we only collect a small number of physical event descriptions in the first stage. In the second stage, we augment these data by training neural event description models based on the ground truth physical trajectories of objects and ask human annotators to filter incorrect or hard-to-interpret descriptions.

Finally, we benchmark several machine learning models on CLEVRER-Humans. We demonstrate that our dataset is challenging, in particular, due to the diversity of event descriptions and the challenge of data-efficient learning. The development of CLEVRER-Humans can be beneficial to both machine learning and cognitive science communities. From the machine learning perspective, CLEVRER-Humans posits a combined challenge of natural language understanding, physical grounding of language, and causal reasoning in physical scenes. It is also a stimulus set for cognitive science studies of human physical event perception, causal judgment, and description [6, 11].

| Dataset | Video | Question Answering | Diagnostic Annotation | Natural Language Events | Causal Reasoning | Human Causal Judgements |
|---|---|---|---|---|---|---|
| CLEVR [12] | - | ✓ | ✓ | - | - | - |
| MovieQA [13] | ✓ | ✓ | - | ✓ | - | - |
| TGIF-QA [14] | ✓ | ✓ | - | ✓ | - | - |
| TVQA+ [15] | ✓ | ✓ | - | ✓ | - | - |
| AGQA [16] | ✓ | ✓ | - | ✓ | - | - |
| IntPhys [17] | ✓ | - | ✓ | - | ✓ | - |
| Galileo [18] | ✓ | - | ✓ | - | ✓ | - |
| PHYRE [19] | ✓ | - | ✓ | - | ✓ | - |
| CATER [8] | ✓ | ✓ | ✓ | - | ✓ | - |
| CoPhy [20] | ✓ | - | ✓ | - | ✓ | - |
| CRAFT [10] | ✓ | ✓ | ✓ | - | ✓ | - |
| CLEVRER [7] | ✓ | ✓ | ✓ | - | ✓ | - |
| ComPhy [21] | ✓ | ✓ | ✓ | - | ✓ | - |
| CLEVRER-Humans | ✓ | ✓ | ✓ | ✓ | ✓ | ✓ |

Table 1: Comparison between CLEVRER-Humans with other visual reasoning benchmarks. Our dataset is the only dataset of natural language descriptions of physical events and human judgments.

## 2 Related work

**Physical and causal reasoning.** Our dataset is closely related to benchmarks on physical reasoning tasks. In general, these benchmarks can be categorized into three groups based on their evaluation protocol. First, datasets such as IntPhys [17], Galileo [18], and CoPhy [20] focus on making counterfactual or hypothetical predictions of physical events. Second, benchmarks including PHYRE [19] and ESPRIT [22] focus on an "inverse" problem of generating initial conditions that leads to a specific goal state. The third group of benchmarks, where our dataset CLEVRER-Humans also falls into, focuses on assessing machine learning models through a natural language interface, such as CATER [8], CLEVRER [7], CRAFT [10], and ComPhy [21]. While these datasets focus on other aspects of physical and causal reasoning, such as physical property inference [18] and few-shot learning [21], their causal relationships and event descriptions are all generated by synthetic rules. By contrast, our dataset contains human-annotated physical event descriptions and causal relationships.

**Human causal cognition.** Understanding how humans perceive and reason about causal relationships has been a long-standing problem in cognitive science [9]. A number of theories have been proposed to model human causal cognition, such as Force Dynamics [1, 2], Mental Models [23], Causal Models [24, 25], and Counterfactual Simulation [26, 27]. These theories have a variety of aims, including causal induction, causal attribution, and grounding causal language such as "cause", "prevent", "help", "enable", and "allow". In this work, we study the relatively neutral causal connective "because", together with freely-generated descriptions of physical events.

Physical causation, in particular, has received much attention with rich experimental and theoretical literature. Among theories of physical causation judgements: Conserved Quantity Theory [28] predicts causal relations by inspecting how conserved quantities (e.g., momentum) transferred between events. Force Dynamics Theory [FDT; 1, 2] focuses on analyzing the mechanism of causation when bodies interact, using force vectors to represent the interaction between objects, and predicting judgments based on the direction and length of these vectors. Counterfactual Simulation Theory [CSM; 26, 27] leverages an intuitive physics model to make counterfactual judgements (A causes B if A's occurrence makes a difference to B's occurrence). These theories often agree on predicted causal judgements, and are very often different from the simple heuristic used to annotate the "responsible for" relation of CLEVERER. There are also critical differences in predictions, particularly for complex events (where it is not always clear how to apply a given theory). Thus it is necessary to collect human judgements of the causal relation between events; in future work, it will be interesting to compare machine learning models derived from our data to psychological theories of physical causation.

**Video question answering.** The primary task we study is to answer questions about videos of physical scenes. Unlike benchmarks that focus on multi-modal learning in real-world videos about human-

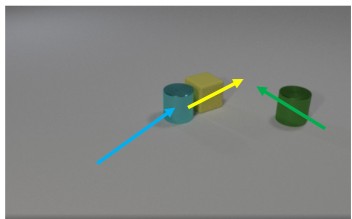

**Question**: What is responsible for the rubber cube's colliding with the green object?
**Choice**: The presence of the cyan object.
**CLEVRER-Heuristic**: Correct
**Human**: Wrong
Explanation: Humans may think the "presence" is not the direct cause. Instead, the collision between the cyan object is a more direct cause.

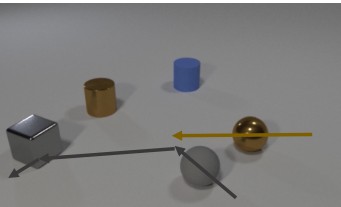

**Question**: What is responsible for the rubber sphere's exiting the scene?
**Choice**: The presence of the metal sphere.
**CLEVRER-Heuristic**: Correct
**Human**: Wrong
**Explanation**: Even without the presence of metal sphere, the rubber sphere will still exit the scene. Also, before exiting the scene, the rubber sphere collides with the grey cube. Thus, the choice is not the cause.

Figure 2: This are two examples showing difference of human causal judgment and CLEVRER's heuristic causal relation. The arrows in the image represent the moving direction of objects of interest.

| P(X\|Y) | Y = CLEVRER | Y = Counterfactual | Y = Human |
|---|---|---|---|
| X = CLEVRER | 1.00 | 0.74 | 0.96 |
| X = Counterfactual | 0.89 | 1.00 | 0.54 |
| X = Human | 0.62 | 0.61 | 1.00 |

Table 2: Comparison between different heuristics-generated causal labels and human labels, on a sampled subset of CLEVRER [7]. The entry P(X|Y) denotes the fraction of event relations that are annotated as causal by protocol X given that the relations are annotated as causal by protocol Y.

human and human-object interactions [13, 14, 15, 16], we focus on understanding physical events and their causal relationships. We summarize the key factors that differentiate our dataset from others in Table 1. Among them, CLEVRER-Humans is the only dataset that contains human-annotated physical events and causal judgements.

**Video-conditioned text generation.** Our dataset is built on recent advances in using deep neural networks to generate natural language descriptions of videos [29, 30, 31, 32]. A common approach to generating video event descriptions is to first apply per-frame neural network encoders to obtain per-frame features, and aggregate them over time. Popular aggregation algorithms include average pooling [33], recurrent neural networks [34], and attention mechanisms [35]. Our model for event description generation falls into the third group, and leverages temporal attention in generation.

## 3 CLEVRER-Humans

Our dataset is based on the videos from the CLEVRER dataset [7]. Each video contains at most 5 colliding objects moving on a single plane in 3D. The objects have 3 different shapes (cylinder, cube, and sphere), 8 colors, and 2 materials (rubber and material). The questions in CLEVRER consists of four types: descriptive ("what color"), explanatory ("what's responsible for"), predictive ("what will happen next"), and counterfactual ("what if"). In this paper, we particularly focus on explanatory questions, which evaluate machine reasoning about causal relationships. Explanatory questions query the causal structure of a video by asking for the cause of an event, providing multiple choices. There are 3 types of events defined in CLEVRER's explanatory questions: entering the scene, exiting the scene, and colliding with another object. In CLEVRER, the event descriptions are generated by pre-defined templates.

### 3.1 Delving into Causal Relation Annotations

Before introducing our new dataset CLEVRER-Humans, we start from investigating the divergence between human labels and the labels generated by the original CLEVRER dataset using heuristics. In this section, we will be comparing three labelling protocols on a subset of 50 videos from the CLEVRER dataset. The first method (CLEVRER) uses the heuristic rules defined in the original CLEVRER dataset to predict causal relations between each pair of events. Specifically, if event A

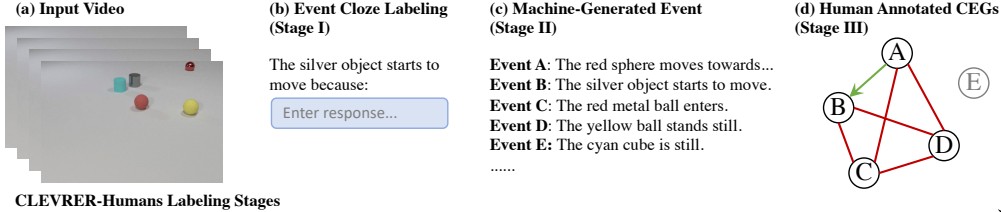

**CLEVRER-Humans Labeling Stages**

(a) **Input Video**

(b) **Event Cloze Labeling (Stage I)**

The silver object starts to move because:

Enter response...

(c) **Machine-Generated Event (Stage II)**

**Event A**: The red sphere moves towards...
**Event B**: The silver object starts to move.
**Event C**: The red metal ball enters.
**Event D**: The yellow ball stands still.
**Event E:** The cyan cube is still.
......

(d) **Human Annotated CEGs (Stage III)**

Figure 3: The overall labeling pipeline of CLEVRER-Humans. (a) Starting from input videos, (b) we use a event cloze task to collect a small number of human-written event descriptions about videos (Stage I). (c) Next, we train neural event description generators to augment all videos with a collection of events (Stage II). (d) Finally, human annotators label the correctness of all generated events (in this case, event E is incorrect and thus the node is dropped) as well as their causal relations (Stage III).

happens before event B, and they share at least one object, we say event A is a cause of event B[†]. The second method (Counterfactual) uses counterfactual intervention to derive the causal relationship. Specifically, we say event A causes event B if event A happens before event B and event B happens even if we remove all relevant objects in event A from the scene (except for the objects in event B). We compare these two methods with human labeled causal relations.

We summarize the resulting statistics in Table 2. There is a considerable difference between heuristic labels and human judgments. Compared to counterfactual intervention, the CLEVRER heuristic is a closer approximation to human judgment. We think this is partially because we have kept the term "responsible for" used by CLEVRER other than "cause" in the human labelling interface. Fig. 2 provides two illustrative cases with explanations. We also provide detailed analysis on the effect of binarization thresholds and compositions of heuristics in the supplementary material.

## 3.2 Augment CLEVRER with Human Annotations

The key representation we will be using to augment CLEVRER is the Causal Event Graph (CEG), which has been adopted in similar datasets on video causal reasoning such as CRAFT [10] and video reasoning [36]. Each CEG $\mathcal{G} = (\mathcal{V}, \mathcal{E})$ is a graph structure of events and their causal relationships in a video. Each node $v \in \mathcal{V}$ is a natural language description of a physical event in the video, and each directed edge $(v_1, v_2) \in \mathcal{E}$ has a label in {*positive, negative, unlabelled*}[‡]. A positive label for $(v_1, v_2)$ denotes that event $v_1$ is a cause of event $v_2$. CEGs directly enable downstream tasks such as physical and causal event generation, and causal relationship classification. Furthermore, based on the dense graph representation $\mathcal{G}$, we can generate questions about the causal relations, e.g., "Which event is responsible for the red cube moving?"

There are two desiderata for CEG annotations: diversity in event descriptions (nodes) and density in edge labels. Ideally, we want to ask human annotators to label all events in a video and relationships for all pairs of events. However, this is very time- and budget-consuming. Therefore, in practice, we use a three-stage annotation pipeline, as illustrated in Fig. 3. The first stage focuses on collecting human-written event descriptions using event cloze tasks, but only for a small number of videos. In the second stage, we augment the data for all videos using neural event description generators trained on the data collected from the first stage. In the third stage, we condense CEGs by collecting binary causal relation labels for all pairs of events from humans. All data are collected using the Mechanical Turk (MTurk) platform.

## 3.3 Stage I: Event Cloze

In the first stage, we use an event cloze task to collect human-written event descriptions. Cloze tests have been employed in various natural language processing (NLP) domains, such as narrative

---

[†]For simplicity, we use the word "cause" in this section. In the original CLEVRER dataset, such a relationship is formally defined as "*A is responsible for B.*"

[‡]Note that CEGs are generally different from causal graphical models because edges in CEGs are manually labelled and they reflect human judgement. Thus, they may not comply inference rules in graphical models.

| Object State | Relative Position | Collision |
|---|---|---|
| Object A comes/moves/is hurled from some direction. | Object A is in the path of object B. | Object A is hit/struck by object B |
| Object A is thrown/pushed from some direction. | Object A is aimed at object B. | Object A hits/collides with/bumps into/runs into object B |
| Object A changes direction (towards some direction) | Object A moves together with object B | Object A is avoided hitting object B. |
| Object A stops moving | | Object A pushes object B into object C. |

Table 3: Event examples collected in the first stage, categorized into single-object states, pairwise relative positions, and pairwise collision events.

cloze [37], reading comprehension [38] and story-telling [39]. In contrast to these existing work, we develop a novel *iterative* data collection procedure as shown in Figure 4. For each video, we initialize its CEG with a single node using an event description from CLEVRER. Then we iteratively sample a node in the CEG, and ask humans to annotate a cause event or an effect event of the sampled event.

Specifically, the MTurk interface contains a single video and an incomplete sentence about two events in the video, connected by a "because" discourse marker: "_____ because the yellow cylinder collides with the purple square." In this case, the human-written event is an effect of the event specified in the sentence. Similarly, we use the template "Event A happens because _____." to collect cause events. Most notably, the newly collected events will be used as the seeding event in the next iteration. This progressive design thus improves the diversity and coverage of the collected event descriptions.

In this stage, we select 100 videos from the CLEVRER training set with the largest number of collisions since we want to work with videos with especially rich causal relationships. In total, we obtain 1,724 event descriptions from 100 CLEVRER videos through crowdsourcing. We recognize three major types of events in the stage I dataset as shown in Table 3.

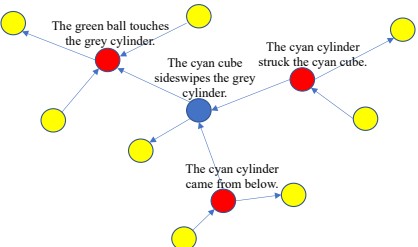

Figure 4: We use a novel iterative data collection procedure to collect CEGs on MTurk. Starting from a single node (iteration 0, blue), we iteratively sample nodes in the current CEG and collect either a cause or an effect event, and add this new node to the CEG. Red: nodes added in the first iteration; yellow: nodes added in the second iteration.

### 3.4 Stage II: Trajectory-based Event Description Generation

In the second stage, we leverage the manually written event descriptions to train neural description generators and augment the event description data for all videos. Our overall pipeline is shown in Fig. 5. Given the input video, it uses two branches to generate single-object events and pairwise events. The generated event descriptions will be sent to a post-processing module. It is important to note that the design choices in this stage focus on the *coverage* (i.e., we want to recover as many events from the input video as possible). All generated descriptions will be filtered again by MTurkers.

**Trajectory representation.** Instead of working with pixels, all event generation modules take symbolic representations of object trajectories as their inputs. Specifically, each video is represented by a set of trajectories $\{s_{i,t}\}$, where $s_{i,t}$ is the state of object $i$ at timestep $t$. The state contains the color, shape, and material properties of object $i$, as well as its 3D position, velocity, and angular velocity at time step $t$. Since the CLEVRER videos are generated using simulated physical engines, this information can be directly obtained. Compared with images, trajectory-based inputs are lower-dimensional, and empirically yield significantly better results than pixel-based inputs when trained on a limited amount of data.

**Data pre-processing.** To associate human-written event descriptions with a single object or a pair of objects, we first run noun-phrase detection methods[§] on all descriptions. Next, we use all concept

---

[§]In particular, we used the tools from spaCy: https://spacy.io/.

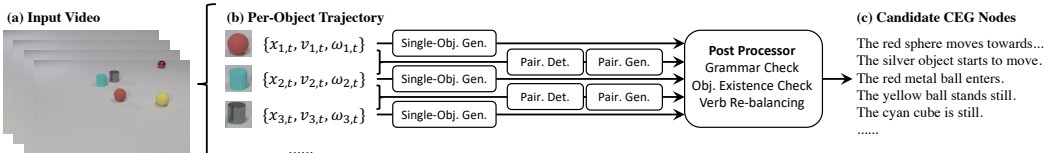

Figure 5: The neural event description generation pipeline (Stage II). (a) Given the input video, (b) we first extract per-object trajectories, composed of their attributes, positions, velocities, and angular velocities. For each object, we use a single-object event generator to sample event descriptions. For each pair of objects, we use a cascaded generator composed of a rule-based event detector and a neural pairwise generator. All generated events will pass a post-processing unit composed of three stages: grammar check, object existence check, and verb re-balancing. (c) The final product of the pipeline is a candidate node set of the CEG, which will be further annotated by humans in Stage III, the CEG condensation stage.

words in the detected noun phrases to filter the objects being referred to. The concept vocabulary contains colors, shapes, and materials, and is manually annotated. Based on the detected objects, we categorize each description into single-object (pairwise) events, and associate them with the corresponding object (pair of objects).

**Single-Object event description.** The single-object event generator is a GRU-based sequence-to-sequence model with attention [40]. It takes the trajectory of a single object throughout the video as input and generates natural language descriptions of events associated with this object. This module is trained on all of the 740 single-object descriptions from stage I.

**Pairwise event description.** Unlike single-object events, an important feature of pairwise events is *sparsity*. Specifically, for most pairs of objects in the video, there is no event associated with them. However, because we don't have *negative* data in pairwise events (i.e., there are no descriptions such as "*object A and object B do not have interactions.*"), direct training of pairwise event generators will yield a lot of false-positive detection during test time.

To address this issue, we employ a rule-based object pair filter before the event generation process. Concretely, the event detector takes in the trajectories of a pair of objects and outputs whether an event occurs for the input pair. It first detects the longest consecutive (increasing/decreasing) sequence of object positions to segment the input trajectories. Then it processes each segment based on the physical properties in the trajectory. Based on manual inspection of stage-I data, we use rules to detect three types of events: object approaching, collision, and moving together. This is a simple but effective pre-processing step. By choosing the threshold, our detector has a recall rate of 99.2% on a small manually labeled dataset consisting of 100 videos. On the same split of 100 videos, the event detector improves the pairwise event description accuracy, labeled by human annotators in stage III, from 80.6% to 87.2%. For all pairs selected by the pairwise event detector, we concatenate their trajectories and feed them into a separate GRU-based sequence-to-sequence model. Similar to the single-object generator, this model is trained on all of the 984 pairwise descriptions from stage I.

**Post-processing.** We employ three post-processing steps to ensure the quality and diversity of generated event descriptions: grammar checking, object existence checking, and verb re-balancing. We first generate a large set of event descriptions (top-10 likely sequences per object and per pair). Next, we perform a grammar check using hand-crafted rules to filter common grammar errors such as missing verbs or missing verb arguments. Then, we detect noun phrases in the event descriptions and use concept words to select the object being referred to. We drop event descriptions that refer to objects that do not appear in the video. Finally, we notice that due to the sampling strategy, descriptions with the highest probabilities are not diverse: for example, for pairwise events, most highest-probability descriptions are about object collisions. Thus, we re-balance the data distribution based on the main verb of the descriptions. Specifically, we match this distribution with the human-written descriptions in stage I. For each video, we randomly sample at most 10 event descriptions.

### 3.5 Stage III: CEG Condensation

Finally, based on event descriptions generated in the second stage, we build dense CEGs using MTurk. In contrast to the first stage, where we use an event cloze task to collect natural language descriptions, in this stage, we focus solely on edge labeling: classifying whether two events have a causal relation.

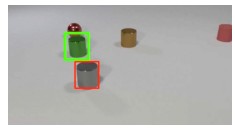

| **Question 1:** Choose one. Here "incorrect" means both grammatically or factually. | **Question 2:** How much is **event A** responsible for **event B**? |
|---|---|
| **Choice 1:** Event A is incorrect | **Choice 1:** not responsible at all |
| **Choice 2:** Event B is incorrect | **Choice 2:** a bit responsible |
| **Choice 3:** Both event A and B are incorrect | **Choice 3:** moderately responsible |
| **Choice 4:** Both event A and event B are correct | **Choice 4:** quite responsible |
| | **Choice 5:** extremely responsible |

**Event A:** the green cylinder moves left.
**Event B:** the red cylinder bumped into the gray cylinder.

Figure 6: The MTurk interface illustration for the second stage. The MTurker sees the input video with objects mentioned in the events highlighted with colored bounding boxes. The MTurker answers two questions about the video: 1) whether event descriptions are correct and 2) a graded judgement of whether event A is responsible for event B. We use the same wording choice "be responsible for" to be consistent with the original CLEVRER dataset [7].

Concretely, we use an interface depicted in Fig. 6. The annotator is presented with a video, with objects being referred to in event descriptions highlighted in different colors. The annotator needs to answer two questions. The first question asks whether two event descriptions are interpretable (e.g., grammatically correct) and correct (i.e., they happen in the video). If the annotator labels both descriptions correct, they need to answer a second question, asking for a graded judgment (score 1-5) about whether the first event is a cause of the second event.

Based on the responses, we drop CEG nodes where a majority of the annotators label incorrect. The labeled edges have a score ranging from 1 to 5. We binarize the labels with a threshold of 4. That is, all edges with scores greater than or equal to 4 are considered positive.

## 3.6 QA conversion

To be compatible with the existing dataset CLEVRER, we further convert the CEGs to multiple-choice question-answer pairs. For every node $v$ in the CEG $\mathcal{G}$ with an adequate amount of positive parents (i.e. "causes"), we uniformly sample the number of correct choices in the question. We convert it to a question "Which of the following is responsible for $A$?" and sample the set of positive and negative answers following the practice of CLEVRER. This ensures that the distribution of positive and negative candidates is balanced, as in the original CLEVRER dataset. We use the same train, validation, and test splits as CLEVRER.

## 3.7 Dataset Statistics

Overall, in CLEVRER-Humans, we retrieved 1108 videos with 8581 descriptions and 21167 event relationship annotations, after dropping empty CEGs. Based on the CEGs, we generate 1076 question-answer pairs. We will discuss the statistics of CLEVRER-Humans in detail in the supplementary material. In this section, we summarize four important features of CLEVRER-Humans.

First, CLEVRER-Humans contains dense annotations of causal relations between physical events. The average number of CEG nodes is $4.71$, and the average number of labeled edges is $12.7$. These dense annotations of CEGs form the rich and complicated causal structures in our dataset. Second, CLEVRER-Humans offers diverse free-form language descriptions while retaining balances in object properties. CLEVRER-Humans has a vocabulary length of size 219, which is much greater than CLEVRER (82). Next, most importantly, CLEVRER-Humans engage in a variety of physical events for causal reasoning tasks. In particular, CLEVRER-Humans contains 31 distinct verbs, and verbs are used in different tenses. In comparison, the original CLEVRER dataset contains only three event types (and verbs): enter, exit, and collide. Therefore, CLEVRER-Humans significantly improves diversity and brings in a challenge for machines to recognize and ground these events in practice. Finally, CLEVRER-Humans' annotation reflects the subjective judgment of causality in physical events. CLEVRER-Humans offers 5 choices when asking MTurkers to label the causality level. The average score is 2.37. Note that this distribution is skewed towards lower scores. This reflects the fact that most event pairs do not have causal relationships. Finally, although we have binarized the edge labels for the sake of consistency with CLEVRER, the raw score-based judgment can be potentially helpful in other tasks, such as cognitive science studies.

Therefore, we can conclude that CLEVRER-Humans is a high-quality causal relation dataset with significantly more diverse event types and language descriptions than CLEVRER.

# 4 Experiments

In this section, we present the question-answering results of a collection of baseline methods on our new dataset CLEVRER-Humans. Since we have adapted the same input-output interface as the original CLEVRER dataset [7], most methods that are applicable to CLEVRER can also be applied to our dataset. In particular, we compare the following representative ones and highlight the challenge of our diverse and human-annotated physical and causal reasoning task.

## 4.1 Methods

**Per-option best guess.** The most basic baseline, per-option random guess (Guess) classifies each option with the most-frequent answer ("No" in our dataset).

**Language-only models.** We use language-only models to test potential language biases in the dataset. Specifically, we use an LSTM encoder [41] to encode the natural language question and the option, concatenate the last hidden state of both sequences, and apply a linear layer for binary classification.

**Program-based video QA models.** We also compare our model with NS-DR [7] and VRDP [42], two state-of-the-art video reasoning models based on program representations of questions. In particular, both methods leverage a semantic parser to parse questions and options into symbolic programs with hierarchical structures and execute the program based on the abstract video representation extracted by an object property network (NS-DR), or neuro-symbolic concept learning modules (VRDP). We train the semantic parser on the original CLEVRER dataset using the ground truth program annotations and use it to parse newly-annotated CLEVRER-Humans questions.

**End-to-end video QA models.** We also evaluate three additional end-to-end video QA models: CNN+LSTM, CNN+BERT, and ALOE [43]. In CNN+LSTM, we use CNNs to encode each frame into latent vectors and use two separate LSTMs to encode the video sequence and questions, respectively. The output of both video LSTM and the question LSTM are concatenated and fed into another linear layer to make binary classifications. The CNN+BERT model uses a similar architecture except that we replace the LSTM encoder with a pretrained BERT encoder [44]. ALOE [43] is the state-of-the-art video reasoning model on CLEVRER. It is based on the Transformer architecture [45]. It uses MoNET [46] to extract an object-centric representation of videos, uses a multi-modal transformer to encode both object features and questions, and predicts the binary label.

## 4.2 Results

Our results are summarized in Table 4. For both program-based models, since we do not have program annotations for CLEVRER-Humans questions and options, they are trained only on the original CLEVRER dataset and tested on CLEVRER-Humans. For both end-to-end models, we compare three alternatives: trained on CLEVRER, trained on CLEVRER-Humans, and pretrained on CLEVRER and then finetuned on CLEVRER-Humans. For most models that are pretrained, we used the checkpoints released by the authors. Thus, we are unable to compute confidence intervals.

Overall, all methods perform poorly on our dataset CLEVRER-Humans, especially when compared with the best guess model. In general, these results highlight three distinctive challenges of CLEVRER-Humans. First, the diversity of events: the vocabulary of CLEVRER-Humans questions and options are significantly richer than the original CLEVRER dataset. As a result, the performance of models pretrained on CLEVRER significantly drops (compared to their CLEVRER performances). In particular, the original CLEVRER dataset only has a vocabulary size of 82, while CLEVRER-Humans has a vocabulary size of 219. To directly apply pretrained models, we have to encode a large portion of textual inputs as "out-of-vocabulary." Furthermore, by comparing CNN+LSTM and CNN+BERT, we see that using pretrained language models is not necessarily helpful for generalization to unseen events. The second challenge that accounts for the performance gap between the original CLEVRER and our dataset is human-annotated causal judgments: we do not see a significant difference between models trained from scratch on CLEVRER-Humans and the ones pretrained on CLEVRER in terms of the final performance, which empirically suggests the gap between causal relation labels between two datasets. A third challenge that arises from training-from-scratch or finetuning is the limited size of the CLEVRER-Humans training set. Recall that our dataset is directly comparable with the "explanatory questions" category of CLEVRER. CLEVRER contains 122,461 explanatory question pairs, while CLEVRER-Humans contains 1076 pairs. As a result, we

| Model | Training | CLEVRER | | CLEVRER-Humans | |
|---|---|---|---|---|---|
| | | Per-Option | Per-Ques. | Per-Option | Per-Ques. |
| Best Guess | N/A | 50.2 | 16.5 | 50.7 | 31.6 |
| Lang-Only | Scratch | 59.7 | 13.6 | 51.9 ($\pm$ 1.09) | 30.4 ($\pm$ 1.90) |
| NS-DR [7] | Pretrain | 87.6 | 79.6 | 51.0 | 32.0 |
| VRDP [47] | Pretrain | 96.3 | 91.9 | 50.9 | 31.6 |
| CNN+LSTM | Pretrain | 62.0 | 17.5 | 50.3 | 30.0 |
| CNN+LSTM | Scratch | N/A[†] | N/A[†] | 51.7 ($\pm$ 0.64) | **34.2** ($\pm$ 1.69) |
| CNN+LSTM | Pretrain+Finetune | 62.0 | 17.5 | 51.5 ($\pm$ 2.35) | 30.8 ($\pm$ 0.69) |
| CNN+BERT | Pretrain | 55.1 | 11.5 | 52.9 | 32.0 |
| CNN+BERT | Scratch | N/A[†] | N/A[†] | 52.0 ($\pm$ 2.34) | 30.2 ($\pm$ 2.41) |
| CNN+BERT | Pretrain+Finetune | N/A[†] | N/A[†] | 50.1 ($\pm$ 0.68) | 30.4 ($\pm$ 3.09) |
| ALOE [43] | Pretrain | **98.5** | **96.0** | **54.0** | 26.9 |
| ALOE [43] | Scratch | N/A[†] | N/A[†] | 51.8 ($\pm$ 1.00) | 31.7 ($\pm$ 0.79) |
| ALOE [43] | Pretrain+Finetune | **98.5** | **96.0** | 52.7 ($\pm$ 1.36) | 32.1 ($\pm$ 1.36) |
| Human | N/A | N/A | N/A | 84.5 | 71.4 |

Table 4: Test performance of different models on both the original CLEVRER dataset and our new CLEVRER-Humans dataset. The training column denotes the training schema for different models (in percentage, and the $\pm$ sign shows the 95% confidence interval computed across 5 different runs). We compare both per-option accuracy and per-question accuracy, following the original paper [7]. Note that the number of options per question is ~4 for CLEVRER and 2 for CLEVRER-Humans. N/A marker[†]: models trained only on CLEVRER-Humans.

observe extreme training-set overfitting for large models such as ALOE. While as training goes on, the training accuracy keeps increasing to 70%, the per-option test accuracy plateaus at 53.7% at a very early stage. The combined challenge of diversity and data-efficient learning can be potentially addressed by enabling better transfer learning from large pre-trained multi-modal models such as CLIP [48], and better physics-informed models [49]. We also include human performance on the test set. Participants with full professional or higher proficiency in English are asked to evaluate the results on 50 videos from the test set. The participants also provide the percentage of descriptions that appear natural to them, which is 90.0% on average.

## 5  Conclusion

We have presented CLEVRER-Humans, the first video reasoning dataset of human-annotated physical event descriptions and their causal relations. CLEVRER-Humans introduces a unique and important challenge of combined physical scene understanding, natural language understanding, and causal reasoning. Due to its limited size, CLEVRER-Humans should be primarily used for zero-shot evaluation or few-shot training. Our preliminary results on question-answering illustrate the challenge of interpreting diverse human-written event descriptions, making human-like causal judgments, and data-efficient learning. To collect the dataset within a reasonable budget, we introduce two important techniques for dataset collection: iterative cloze-based annotation of event descriptions, and hybrid event description generation using neural sequence-to-sequence models. Both techniques can be applied to elicit data for any similar relation (such as "before" and "supports"). An exciting challenge will be extending these methods to naturalistic videos, complex events, and ambiguous relations.

Throughout the evaluations in this paper, we have been focusing on a binarized version of the causal reasoning task. However, it is important that computational models can make graded judgements as humans do. While existing literature has studied graded causal reasoning in text domains [50, 51] and static images [52], we believe an important direction is to build better models that perform graded causal reasoning in dynamic videos.

**Acknowledgements.** This work is partly supported by the Stanford Institute for Human-Centered AI (HAI), the Samsung Global Research Outreach (GRO) Program, ONR MURI N00014-22-1-2740, and Adobe, Amazon, Analog, Bosch, IBM, JPMC, Meta, and Salesforce.

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
