# OpenReview forum: "CLEVRER-Humans: Describing Physical and Causal Events the Human Way"
_NeurIPS.cc/2022/Track/Datasets_and_Benchmarks — NeurIPS 2022 Datasets and Benchmarks _

### Official Review · Reviewer_5aig · 2022-07-05
**human judgements annoation over CLEVRER dataset with NN data augmentation**

**Rating:** 8
**Confidence:** 4

**Strengths:**

It is important for AI to understand physical events. In previous datasets, the description of what happens (events) and their causal relations are based generated manually by heuristics. But these annotations might not be precisely how humans understand physical events. This work aims to bridge the gap by collecting human annotations. This is the strength of this work.


(UPDATE: I moved my score from 4-reject to 8-clear accept after the author made addressed the core weakness - that is, how is like of this gap between heuristics. It is an important question worth the field's notice, and their dataset provides a good benchmark for it, so I give an 8.)

**Weaknesses:**

However, one central problem they aim to address - the difference between human understanding and heuristic generation of causal relationship and it’s importance - is not justified and presented clearly in the paper. Though this paper seems good elsewhere, if this fundamental flaw is not fixed, the whole benchmark could turned out to be meaningless and mislead other people’s research direction. This is why I give a rejection for now.

Specifically:

1, How is humans' causal judgment different from the template-generated one in CLEVRER? Why and how do they matter?
There seems to have no comparison and analysis. The reference paper given for this general topic does not address this issue directly.

2, Also, why would humans have different causality scores on an event? What makes human uncertain? Do these different scale correspond to different types of causality or just because of lack of information? Does it matter?


Also notably:

3, if the point is to collect human diversity, doesn't it defeat the purpose to use NN to generate descriptions? I know it's slow and costly to let all tasks to be annotated by human, but this execution difficulty doesn't justify whether the NN method is valid. Even if you did some verb distribution post-processing to alleviate this issue, but how can you prove they are sufficiently diverse and human alike? Humans might just come up with some novel way to look at the event and describe differently.



**Additional Feedback:**

One more question: you retrieved 1108 videos with 8581 descriptions and 21167 event, but why can you generate only 1076 question-answer pairs?


**Clarity:**

The content included in this paper is clearly written, but as mentioned above, some key justifications and analyses are missing. This caused trouble in understanding some key contribution of this work.

**Correctness:**

As stated in the paper, there're only 1076 question answer pairs, so the dataset size is very limited to enable meaningful algorithm training (at least NN based). So it isn't clear to me that the evaluation is meaningful.

**Documentation:**

There is a json interface.
It would be greater if the author can provide the code they use to run the experiments.

**Ethics:**

There is no ethical issue in this paper that i can think of.

**Relation To Prior Work:**

The relation to prior work is clear.

**Summary And Contributions:**

This dataset collects human judgement annotations over events and causal relations of a previously existing physical events dataset called CLEVRER. The aim is to address the two issues in CLEVRER: the diversity issue in natural language description and the causal judgement discrepancy between human and heuristic-based generations. It collects 1724 events descriptions from 100 CLEVRER videos from humans, and then us them to train some neural networks to produce more human-alike annotations, in total to 1108 videos with 8581 descriptions and 21167 event relationship annotations, and generated 1076 question-answer pairs.

---

> ### Author Response · Authors · 2022-08-21
> **Response to Reviewer 5aig**
>
>
> **Q1:** Difference between human understanding and heuristic generation.
>
> **A1:** Thanks for the suggestion. We have included additional human experiments to evaluate the divergence.
>
> Specifically, we compare human answers to the original CLEVRER questions with two commonly-used heuristics for automatic generation of the answer. The method Heuristic uses the original CLEVRER heuristic, where event A is directly responsible for event B if A happens before B and they share a common object. The method Counterfactual-Intervention uses counterfactual inference: we removed objects that appear in A but not in B so that A will not happen in the video, and re-run the simulation. We say event A is responsible for B if event B does not happen without A. The results are visualized in the following three-by-three table. The entry p(X|Y) denotes the fraction of event relations that are annotated as causal by protocol X given that the relations are annotated as causal by protocol Y.
>
> | p(X\|Y)          | Y=CLEVRER | Y=Counterfactual | Y=Human |
> |------------------|-----------|------------------|---------|
> | X=CLEVRER        | 1.00      | 0.74             | 0.96    |
> | X=Counterfactual | 0.89      | 1.00             | 0.54    |
> | X=Human          | 0.62      | 0.61             | 1.00    |
>
> Our experiments reveal that all three sets of answers diverge from each other to some degree. In general, the original CLEVRER-Heuristic agrees with human judgments more. Qualitatively, we found that heuristic-generated labels primarily differ human judgment when the cause is indirect (e.g., the question asks whether the presence of an object is responsible for the collision, rather than the movement of that object), or when the event is not the only cause of another event (e.g., ball A and B will still collide even without the collision between B and C).
>
> **Q2:** Human causality scores and the source of uncertainty.
>
> **A2:** We have included additional examples for human causal judgments, with each causality grade labeled by MTurk labelers (https://sites.google.com/stanford.edu/clevrer-humans/examples). In general, we have found that humans tend to give lower scores to events that are indirect cause of the target event or are only part of the cause (i.e., there are other events that jointly cause the target event to happen). However, we do not see consistent patterns that map between causal scores and specific types of causality. The graded judgments have small effects on the models evaluated in this paper, because we have binarized the causality labels to be consistent with CLEVRER.
>
> **Q3:** Diversity in language.
>
> **A3:** This is a good point. We wish to clarify that our goal is to maximize the usefulness of the dataset to the community given the resources we have. We agree that assuming there is an unlimited budget, we would like to collect a large-scale dataset, with all descriptions directly from humans; in practice, we have to consider the tradeoff between the scale of the dataset and the diversity of the descriptions. In our specific case, we have observed that training models to generate event descriptions (based on symbolic state inputs) is much more data-efficient than building models for making causal judgments. This motivated us to use NNs to replace some of the labeling efforts to scale up the dataset without significantly increasing the cost. As a result, we have decided to collect a smaller set of human labeled event descriptions and put more budget and emphasis on modeling human causal judgments.
>
> Furthermore, we have also released the subset of CEGs and questions that are completely written and labeled by humans. They are all from the validation split and are therefore suitable for evaluation purposes. Thus, dataset users can evaluate on this split to completely remove any artifacts that can be potentially created by machine-generated event descriptions.
>
> **Q4:** Dataset size.
>
> **A4:** When generating QA pairs, we sample from CEGs and balance the number of correct/incorrect answers. For each node in CEGs, we randomly generate the number of correct answers and drop this node if its in-degree is not sufficient. Therefore, the balanced dataset has 1076 question-answer pairs. We also have the unbalanced version with 21,167 event relationship annotations. A reference in number can be CLEVR-Humans, with a training set of ~17,000, (https://cs.stanford.edu/people/jcjohns/iep/).
>
> As for baseline evaluations, we consider this dataset as a benchmark primarily for few-shot finetuning and zero-shot evaluation, rather than training video understanding models from scratch.
>
> **Q5:** Code release.
>
> **A5:** We have released the code for baseline models and event description generator at https://github.com/xyang23/CLEVRER-Humans1.0

---

> > ### Author Response · Authors · 2022-08-26
> > **Looking Forward to Your Response**
> >
> > Dear Reviewer,
> >
> > Thank you again for the constructive reviews, which have helped us greatly improve the quality and clarity of our paper. Based on your suggestions, we have added additional human experiments on the difference between heuristics and human causal judgments. The manuscript has also been updated to include this new experiment and the new analysis of human causal judgments. We hope our response and new results have been able to address your concerns. As we approach the end of the discussion period, please don’t hesitate to let us know if you have any additional questions or comments!
> >
> > Thanks for your time,
> >
> > Authors

---

> > ### Comment · Reviewer_5aig · 2022-08-26
> > **Nice updates, but might still need more experiment adjustments and discussions from psychology field**
> >
> > Thanks for the reply and add-on experiment. Here're some following-up comments.
> >
> > 1. I hope you can give more discussions/illustrations/analyses on the difference between human causal judgment and heuristics in the paper's introduction section.  There should be a lot of work from psychology in this field. It'll be beneficial if not necessary for readers and people who work on your benchmark to better understand the issue they are solving.
> >
> > Your work is essentially motivated by the psychology field (how human works), and you hope to bring this gap to AI people's attention. So it is important to establish the meaningfulness and importance of this issue with enough background discussion -- write several paragraphs of discussions and cite a bunch of works, especially the latest works in the psychology field.  Currently, it seems more like a quick intuitive judgment rather than rigorously diving deep into this issue.
> > Particularly since this work is a benchmark, it is more important to do so since the field's future works rely on it as their measurement of progress. The potential risk is that if there are any wrong assumptions in the problem definition, the whole subfield could be misled and tons of people's time could be wasted. (this is NOT saying it will -- this is just trying to explain the importance and the risk that should have been minimized)
> >
> >
> > 2. The add-on experiment is definitely helpful, but there are multiple issues:
> > -- 1) Did you cover all important heuristics? If not so, how can we feel sure that the gap is between humans and some heuristics or with any heuristics?
> > -- 2) There should be an evaluation of (different) combinations of heuristics because human judgment might use a combination instead of just one.
> > -- 3) There should be comparisons using different thresholds of the causal score on the gap. Currently, human rate causality is based on scores, but you set the threshold as 4. This surely has an effect on the measurement, and it is very important for us to understand its impact.
> >
> > 3. As we mentioned about the threshold, could you explain why you guys set the threshold as 4?
> >
> > 4. Following on comment 2, can you guys release the code for that add-on experiment? So that people can play with different heuristics themselves and understand this human gap issue better. This could also be a thing in itself for the field.
> > And in order to do so, can you guys also release the raw human causal score data, if not already released?

---

> > > ### Author Response · Authors · 2022-08-28
> > > **Thank You for Your Comments! Manuscript Updated and Code Released**
> > >
> > > Thank you for your response.
> > >
> > > **Q1:** More surveys on human causal cognition.
> > >
> > > **A1:** Thanks for your suggestion. We have included two new paragraphs, highlighted in green, in the related work section to describe related backgrounds in cognitive science studies. We have included a representative list of cognitive science models for human causal cognition. Moreover, we have discussed the difference between three theories (Conserved Quantity Theory, Force Dynamics Theory, and Counterfactual Simulation Models) for human's causal judgment in physical scenes. These theories have critical differences in predictions, particularly for complex events (where it is not always clear how to apply a given theory). Thus, we collect human judgements of the causal relation between events. In future work it will be interesting to compare machine learning models derived from our data to psychological theories of physical causation.
> > >
> > > **Q2:** Heuristics vs. human judgment.
> > >
> > > **A2:** Thank you for your suggestions! Our study has included two of the most prevalent heuristics for modeling causal relations in both machine learning and cognitive science. Specifically, our counterfactual heuristics is inspired by counterfactual-based models for human causal judgment in cognitive science [1]. The CLEVRER-Heuristic is the primary heuristic used in relevant machine learning datasets, including CLEVRER and CATER [2]. Another important heuristic is the force dynamics model in cognitive science [3]. However, this model reasons about causal relations of events by reasoning about the interaction force between objects, which does not generalize directly to our setting that includes non-forceful events such as "the presence of X" and 3D torques (e.g., spins and sideswipes).
> > >
> > > Per request, we have included additional studies (Table 1 in the appendix and also below) on combining two heuristics by taking the conjunction and disjunction of them. We do not see significant improvements over using each heuristic alone.
> > >
> > >
> > > |                  | Y=CLEVRER | Y=Counterfactual | Y=CLEVRER and Counterfactual | Y=CLEVRER or Counterfactual |
> > > |------------------|-----------|------------------|---------|--------------------|
> > > | p(Human \| Y)    | 0.62      | 0.61             | 0.23   | 0.34       |
> > > | p(Y \| Human)    | 0.96      | 0.54             | 0.29    | 0.62       |
> > >
> > >
> > > We have also included the ablation study of matching heuristics with different score thresholds (shown in Figure 6 in the appendix). In general, increasing the threshold will decrease the matching score p(Human | Heuristic) but increase p(Heuristic | Human), analog to the relationship between precision and recall in classification tasks. We strongly agree that this figure provides a more holistic understanding of the divergence between humans and heuristics. Meanwhile, we would like to clarify that we are not choosing the threshold for the sake of making the human judgment "closer" to any of the heuristics (or their combination). Thank you again for all the suggestions!
> > >
> > > [1] Tobias Gerstenberg, Noah D. Goodman, David A. Lagnado, and Joshua B. Tenenbaum. "A counterfactual simulation model of causal judgments for physical events." Psychological Review 128, no. 5 (2021): 936.
> > >
> > > [2] Tayfun Ates, M. Ateşoğlu, Çağatay Yiğit, Ilker Kesen, Mert Kobas, Erkut Erdem, Aykut Erdem, Tilbe Goksun, and Deniz Yuret. 2022. CRAFT: A Benchmark for Causal Reasoning About Forces and inTeractions. In Findings of the Association for Computational Linguistics: ACL 2022.
> > >
> > > [3] Phillip Wolff. "Representing causation." Journal of experimental psychology: General 136, no. 1 (2007): 82.
> > >
> > > **Q3:** The threshold for binarizing human judgment.
> > >
> > > **A3:** The threshold-based filtering (>=4) has been chosen manually to focus on causal relationships that human annotators feel more certain about, based on the description of scores: Choice 2: A bit responsible, Choice 3: moderately responsible, Choice 4: quite responsible, and Choice 5: extremely responsible. To better justify this threshold, we ask human subjects who participated in the annotation to choose a threshold from 1-5 if they had to binarize their judgment. The average threshold suggested by the participants is 3.6. Therefore, both 3 and 4 are reasonable choices. Furthermore, it is worth noting that since we are finetuning most models on our dataset, the exact choice of the threshold should have only small effects on model performance as long as the thresholds are consistent across training and testing.
> > >
> > > Finally, as discussed in the conclusion section, we have also released the raw score of human judgments and it is important that computational models can make graded judgments as humans do.
> > >
> > > **Q4:** Code and data release for the new experiment.
> > >
> > > **A4:** Sure! We have released the code and human data to our github: https://github.com/xyang23/CLEVRER-Humans1.0/tree/main/causal-diff.

---

### Official Review · Reviewer_fmo1 · 2022-07-23
**Neat idea, could use more discussion of uncertainty, and fair pay**

**Rating:** 7
**Confidence:** 4

**Strengths:**

 - The authors identify an interesting problem in the field: Many physical causal reasoning datasets have limited natural language annotations and causality annotations that do not effectively capture detail and nuance in the visual data.

-  The authors apply an annotation method used in various works in pure NL to a visual domain

 -  The paper is well-written and has a detailed appendix. The website is informative and easy to navigate. The online annotation interface is very helpful.

**Weaknesses:**

- It is not clear that the CLEVRER-Humans dataset captures meaningful nuances in human causality reasoning that could not be captured by a well-designed heuristics-based annotation approach (which would be more cost-effective and less noisy). The paper also raises some questions about the quality of the produced annotations.

- Figure 6 shows that the new set of descriptions includes significantly more verbs than the original dataset (one of the primary contributions of the new dataset), but there is no evidence in the paper that these verbs successfully capture meaningful details in the visual data. What is the average agreement among annotators w.r.t. verbs? Would humans agree that the verbs chosen by the language model are the most accurate? In the stage 3 annotation process it seems that annotators only have the option to flag a description as technically incorrect, but not to flag a description as unnatural or poorly worded. Therefore, to a reader it may be unclear whether these model-generated descriptions successfully capture nuances of human causal reasoning.

- Furthermore, it is not clear what the descriptions “a bit responsible” vs. “moderately responsible” vs. “quite responsible” mean in the context of physical causality, there is no analysis of this with examples provided.

**Additional Feedback:**

Neat idea :)

**Clarity:**

The paper is mostly well written and easy to follow. However, the dataset creation process is described both in terms of a 2- and 3-stage process (in the introduction, and in the rest of the paper) which is difficult to follow.

**Correctness:**


In the introduction, the authors claim they show that their dataset is challenging because of the diversity of event descriptions and data-efficient learning. However, based on the experiments section it is not clear that these reasons are the primary causes for the low model performance given the very small dataset size and domain shift from the CLEVRER dataset.

Other than this and the points addressed in the “weaknesses” section, the claims presented in the paper seem to be justified and correct.

**Documentation:**

Yes, the dataset is well documented and a datasheet is included. The website interface is very good and includes a full copy of the annotation interface.

**Ethics:**

It is concerning that the Amazon Mechanical Turk annotators were paid approx. $6/hr to annotate the CLEVRER-Humans dataset, which is below the mandated U.S. federal minimum wage.  This is strikingly low: I encourage the authors to recognize this in their article and commit  to better practice in the future.

https://aclanthology.org/www.mt-archive.info/CL-2011-Fort.pdf

**Relation To Prior Work:**

The authors provide a comprehensive overview of the limitations of existing physical and causal reasoning datasets. However, they omit relevant work in other areas of research (for example, cloze tasks in NLP, graphs for visual event reasoning, etc.), which would add helpful context.

As some examples:

“The language of causation”  argues for a new framework that departs from Talmy and later by Wolff.  Does that now mean anyone within this group of collaborating authors believes that literature is no longer worth referencing?

I disagree with the decision to stick with the binarization as used in CLEVERER as the default for this new dataset. Various works advocate for non-binary treatments, like COPA (which requires alternatives):

https://people.ict.usc.edu/~gordon/copa.html (COPA)
https://aclanthology.org/L18-1316/  (visual COPA)

Or the JOCI work, which generated entailing language then had humans similarly filter and label in a graded way, keeping the graded annotations:

https://aclanthology.org/Q17-1027/

Either way, it would be nice to see more discussion on when there was uncertainty by humans in their judgements.


**Summary And Contributions:**

The authors build on the CLEVRER video reasoning dataset by introducing more varied natural language descriptions and human-annotated causal graphs for CLEVRER visual data. The paper proposes a strategy for producing annotations of this nature in a efficient manner using a combination of human annotations and model-generated labels. They illustrate the challenge presented by this dataset by comparing existing causal reasoning models’ performance on it to their performance on the CLEVRER dataset.


[UPDATE: the authors have adequately responded to concerns]

---

> ### Author Response · Authors · 2022-08-21
> **Response to Reviewer fmo1**
>
>
> **Q1:** Comparison with heuristics-based annotations.
>
> **A1:** This is an important question. As also discussed in our general response, we have compared heuristics-based annotations with human annotations. There is a noticeable divergence between two annotations.
>
> Specifically, we compare human answers to the original CLEVRER questions with two commonly-used heuristics for automatic generation of the answer. The method Heuristic uses the original CLEVRER heuristic, where event A is directly responsible for event B if A happens before B and they share a common object. The method Counterfactual-Intervention uses counterfactual inference: we removed objects that appear in A but not in B so that A will not happen in the video, and re-run the simulation. We say event A is responsible for B if event B does not happen without A. The results are visualized in the following three-by-three table. The entry p(X|Y) denotes the fraction of event relations that are annotated as causal by protocol X given that the relations are annotated as causal by protocol Y.
>
> | p(X\|Y)          | Y=CLEVRER | Y=Counterfactual | Y=Human |
> |------------------|-----------|------------------|---------|
> | X=CLEVRER        | 1.00      | 0.74             | 0.96    |
> | X=Counterfactual | 0.89      | 1.00             | 0.54    |
> | X=Human          | 0.62      | 0.61             | 1.00    |
>
> Our experiments reveal that all three sets of answers diverge from each other to some degree. In general, the original CLEVRER-Heuristic agrees with human judgments more. Qualitatively, we found that heuristic-generated labels primarily differ human judgment when the cause is indirect (e.g., the question asks whether the presence of an object is responsible for the collision, rather than the movement of that object), or when the event is not the only cause of another event (e.g., ball A and B will still collide even without the collision between B and C).
>
> **Q2:** The quality of machine-generated descriptions: whether these model-generated descriptions successfully capture nuances of human causal reasoning.
>
> **A2:** Thanks for the suggestion. We have included an additional human experiment studying the quality of generated event descriptions (other than just true/false.) Specifically, for each event description, we ask participants to annotate if an event description reads natural to them. On an average set of 300 event descriptions from 50 videos, participants agree that 90.0% of the descriptions read natural. We have included this result in the paper.
>
> > Furthermore, it is not clear what the descriptions “a bit responsible” vs. “moderately responsible” vs. “quite responsible” mean in the context of physical causality, there is no analysis of this with examples provided.
>
> **Q3:** Examples of graded judgments.
>
> **A3:** We would like to clarify that in our labeling interface, we provided MTurk annotators with 3 examples labeled by authors as a calibration of the scale. Per request, we have added additional examples of each causality grade labeled by MTurk labelers (https://sites.google.com/stanford.edu/clevrer-humans/examples).
>
> **Q4:** Challenge for models.
> **A4:** We think the poor performance is primarily domain shift. Specifically, based on our human experiments detailed in our General Response Q1, p(Human | CLEVRER-Heuristic) = 0.62. That is, only 62\% of the event pairs that have been labeled as causal in CLEVRER, are labeled as causal by human annotators. This concludes that there is a noticeable divergence between human judgment and heuristics-based labels, which were used for pretraining.
>
> **Q5:** Related work.
>
> **A5:** Thanks! We have included additional references on cloze tasks, and graphs for visual reasoning in Section 3, when we introduce similar concepts. We have also added more literature on causation in Section 1.
>
> **Q6:** Uncertainty in questions.
>
> **A6:** Thanks for the suggestion. We have released the full dataset with graded judgments. They can be used for training machine learning models that generate graded judgments. We have added additional discussions and pointers to the paper you suggested in our revision (Section 5).
>
> **Q7:** Amazon Turk wage.
>
> **A7:** Thanks for the note. We fully agree that this is an important point and wish to clarify. The cloze tests and part of the pairwise causal relationship annotations were completed by users from the U.S., and the pay was above the minimal wage (7.7 dollars/hour). At a later stage of our project, we were unfortunately constrained by the budget available to us and opened the tasks to workers outside the U.S. Thus, overall, our average hourly wage is $6.1. We have expanded the discussion in the supplementary material. Our goal has always been to commit to best practice and offer fair pay to users whenever possible, and we will continue to do so in the future.  Thanks again for raising this point.

---

> > ### Author Response · Authors · 2022-08-26
> > **Looking Forward to Your Response**
> >
> > Dear Reviewer,
> >
> > Thank you again for the constructive reviews, which have helped us greatly improve the quality and clarity of our paper. Based on your suggestions, we have added additional human experiments on the difference between heuristics and human causal judgments, as well as the quality of generated descriptions. The manuscript has also been updated to include these changes, new related work, and MTurk wage details. We hope our response and new results have been able to address your concerns. As we approach the end of the discussion period, please don’t hesitate to let us know if you have any additional questions or comments!
> >
> > Thanks for your time,
> >
> > Authors

---

### Official Review · Reviewer_cqWu · 2022-07-27
**Human-generated causal judgements for physical events**

**Rating:** 7
**Confidence:** 3
**Clarity:** The paper is extremely well written. …

**Strengths:**

- Well-written paper, that presented and clearly described methodology
- Novel graph representation of events (CEGs). An interesting data structure that could be a simplified / more accessible alternative to CGMs
- Using human judgment for causal events
- Well-designed and insightful experimental results showing that the benchmark is challenging to existing methods

**Weaknesses:**

I would like to see an (at least brief) discussion of the similarities and differences between Causal graphical models (CGMs) and the proposed causal event graphs (CEG). For instance, CGMs encode strong (binary) independence and conditional assumptions. In contrast, the questions in stage III allow for a graded type of causal relationship (a bit responsible, moderately responsible, etc.). What is the advantage of this graded way of assessing causality? It would also be interesting to study how consistent subjects are wrt these questions.
I am also a bit worried about the wording here:  For instance, "quite responsible" vs. "moderately responsible". I could imagine that some subjects would rank the word "quite" to be weaker than "moderate". What type of thought process and justifications went into these questions?  A lot hinges on these words as you place the threshold at 4 ("quite"). Personally, I think it would have been better to drop labels for answers with scores 2, 3, and 4 and have 1: not responsible and 5: fully responsible or similar.

Since this is a benchmark paper, I would have liked to have seen more of a discussion of data storage and persistence practices. yes, there is a website and a link, but no DOI, no versioning, etc.

**Additional Feedback:**

Most of my feedback and questions are provided in the "weakness" section.

**Correctness:**

The dataset is constructed in a sound way and the evaluation and experiments are performed correctly.

**Documentation:**

I've mentioned this also in the weaknesses box: there is a lot of room for improvement here. E.g. No hosting, licensing and versioning/maintenance plan.

**Ethics:**

I don't have any ethical concerns. The authors conducted human subject studies but the nature of the problem doesn't raise any ethical questions. It might be nice to provide the payment scheme for the Turkers.

**Relation To Prior Work:**

Prior work is, to the best of my knowledge, fairly discussed. A table contrasts and compares the proposed benchmark with prior work.

**Summary And Contributions:**

The authors use the CLEVRER dataset consisting of video sequences of movable objects and associated physical events and use a sophisticated methodology to annotate these videos with human-generated judgments of causal relationships between the depicted objects. The authors introduce "causal event graphs" which are graph representations of the scenes. Nodes in these graphs are events (e.g., "red sphere moves towards green block" and edges indicate causal or non-causal relationships. To be compatible with existing benchmarks and methods, the authors also provide (question, set of possible answers) pairs.

The data collection methodology consists of three stages.

(I) Collecting human-generated / corrected event descriptions for video sequences.
(II) Event description (single and pairwise) generation using machine learning
(III) Human subjects improve/annotate the event descriptions and conversion of resulting CEGs to QA

---

> ### Author Response · Authors · 2022-08-21
> **Response to Reviewer cqWu**
>
>
> **Q1:** Comparison between causal event graphs and causal graphical models.
>
> **A1:** Thanks for the suggestion. CEGs and CGMs share very similar basic concepts of using representing "causal" relations with edges. The most substantial difference between two representations is that edges in CEGs are manually labeled and graded. The graded judgment allows us to represent richer causal judgments than binary labels. Let's consider that ball A is moving towards ball B, but it collides with ball C which slightly changes its trajectory but ball A still collides with B. In such cases, C may be considered as responsible (e.g., score 4) but probably not the direct cause (e.g., score 5) for A-B collision. By contrast, in CGMs, C will not be considered as a cause for the collision.
>
> **Q2:** The wording "quite" vs. "moderate".
> **A2:** This is a good point. Actually, to avoid confusion, the corresponding levels (in numbers) of different phrases are presented to the annotators in the introduction of the task and also shown together with the phrases.
>
> **Q3:** Design of labels.
>
> **A3:** Thanks for your suggestion. We agree that removing labels for 2, 3, and 4 is an alternative for thresholding. It will have minimal impact on our experimental results, because in the video question-answering dataset we construct, we have binarized the labels. As we have also released the raw annotated score for all edges, follow-up works may explore different ways to binarize the labels or only use part of the labels.
>
> **Q4:** DOI and versioning of the dataset.
>
> **A4:** Thanks! Yes, we will incorporate DOIs and versioning upon the official release of the dataset.
>
> **Q5:** Payment schemes.
>
> **A5:** We have included the payment schemes to MTurkers in the supplementary material (Appendix C)

---

> > ### Author Response · Authors · 2022-08-26
> > **Looking Forward to Your Response**
> >
> > Dear Reviewer,
> >
> > Thank you again for the constructive reviews, which have helped us greatly improve the quality and clarity of our paper. We have updated the paper to include a discussion about causal graphical models. The current Appendix C also contains MTurk payment schemes. We hope our response and new results have been able to address your concerns. As we approach the end of the discussion period, please don’t hesitate to let us know if you have any additional questions or comments!
> >
> > Thanks for your time,
> >
> > Authors

---

> > > ### Comment · Reviewer_cqWu · 2022-08-29
> > > **Response**
> > >
> > > Thanks for your response.
> > >
> > > I would have liked to see a bit more substance in the response to the API, maintenance, versioning, persistence points I raised. It’s easy to dismiss this as something that can be quickly done and added. (Not to say you did, but there is not much evidence you are taking it seriously). For instance, what do you do if you choose to change answer choices? The different versions won’t be comparable anymore. How could others extent the dataset beyond the narrow physics domain? Is there an API available? What are your design choices there? Etc.
> > >
> > > Since I’m also not an expert in human subject studies and psychology, I cannot say much more about the other reviewers’ comments other than that they probably have a point.
> > >
> > > Due to these considerations I’ll lower my score to accept. I still think it is a good paper but have more doubts based on the other reviews.

---

> > > > ### Author Response · Authors · 2022-08-29
> > > > **Thank you for your response**
> > > >
> > > > Dear Reviewer cqWu,
> > > >
> > > > Thank you for your response and detailed explanations on dataset maintenance points. Here we want to briefly respond to your suggestions and summarize our plans for the official release.
> > > >
> > > > We plan to release all data for training and test splits. Our dataset contains two parts: the raw human-annotated scores for CEGs, and the video question-answering annotation created from the CEGs.
> > > >
> > > > Since we are reusing videos from CLEVRER dataset, our data release only contains textual data. For the first part, our data is essentially a graph, where each node is a textual description of an event, and the edge is a 1-5 integer value. Data has been released as a JSON file. We plan to add a JSON schema (https://json-schema.org/) for it. For the second part, our released data has the same format as the original CLEVRER.
> > > >
> > > > We do not foresee important reasons for us to update the existing dataset labels after public release. But we are open to extensions: One possibility is that we will be annotating additional data, in which case the changes will be additive. In this case, we plan to include a unique ID for each event that has been labeled in the dataset, to allow extensions. Another possibility is that we may annotate other relation types (right now we have "responsible for", future work can consider adding "prevent," "enables," etc.). In this case, each edge will contain a list of labels, composed of tuples of (relation type, human judgment score). We plan to switch to this schema in our public release. However, in both cases, it is inevitable the results on old releases will not be comparable. We will always encourage users to use the latest released data (either larger or containing more relation types). After making these two changes, it will be straightforward to extend to other event types (they are just textual descriptions) and relation types.
> > > >
> > > > For DOI and persistent storage, we are working on adapting to the service provided by https://datadryad.org/. They provide both DOI and persistent storage for datasets. Since one institute of some of our authors (Stanford University) already has subscribed services at data dryad, the additional cost will be minimal. Of course, we will also be hosting a copy of the dataset on our institutional servers.
> > > >
> > > > We do not plan to release additional APIs because the data format is already relatively simple and straightforward. However, we have provided examples for simple baselines at https://github.com/xyang23/CLEVRER-Humans1.0/tree/main/models, which include how to read data and feed them into machine learning models. Potential users of our dataset can refer to them.
> > > >
> > > > We deeply appreciate your effort and suggestions on dataset standards!
> > > >
> > > > Authors

---

### Official Review · Reviewer_XFZ1 · 2022-07-27
**Potentially Useful, but Probably Lacking Necessary Quality Checks**

**Rating:** 4
**Confidence:** 3
**Clarity:** Please see weaknesses.

**Strengths:**

Dataset is more diverse (partly resulting from subtleties occurring in human annotated descriptions) than previous counterparts. Pipeline to collect human annotations might be useful/applicable in other works.

**Weaknesses:**

Paper is not exactly self-contained. For example, L31-33, authors say "responsible for" is not same "causes", but did not explain why that's case, they simply mentioned a some references, and left for readers to go through those works and understand.

In my opinion, motivation could have been given more importance. L21-22: "we can use natural language as a lens to evaluate machine understanding of physical events and causal judgments." Any particular why authors want to do this? Is it to generate better human-understandable explanations in case of explainable AI scenario?

Dataset is rather small, and may not be sufficient for training models. Although in vision domain, smaller datasets when curated carefully have shown to be sufficient to train deep networks (YUP++ dataset). This might be indicative of problems in collecting the CH dataset? Please correctness section.

Information on the profile of MTurks is missing. On what basis were the MTurks hired? What's their background? Were they qualified to annotate the dataset? Did they have sufficient knowledge of English language in order to understand the annotation process and questions in the interface and annotate?

Was there any quality checks on human annotations? How many annotators annotated each individual sample? Was any sort of agreement among annotators measured in order to ensure quality checks?

**Additional Feedback:**

N/A.

**Correctness:**

I think quality checks in Stage 3 of human annotation is currently missing. This could lead to wrong annotations, and subsequently the models could be learning from rather ambiguous/noisy data, and that might be the reason for poor performance --- more so than the diversity in the dataset. For example, random baseline is performing more or less on par with trained models (or even better in some cases). Therefore, I don't think the findings are conclusive. I have recommended rejection for this reason. If I am mistaken, please correct me. I will revise my rating based on authors' response.

**Documentation:**

Yes

**Ethics:**

N/A.

**Relation To Prior Work:**

Sufficient, but maybe including prior work on natural language description generation might be better.

**Summary And Contributions:**

This paper proposes CLEVERER-Humans (CH) dataset, where physical events (such as causal relations between objects in case of a collision) are described using natural (human-annotated) language. A multistage pipeline is presented to efficiently collect these annotations. Authors conduct preliminary testing on question-answering task. According to authors. this dataset presents the challenge of interpreting diverse human-written event descriptions, making human-like causal judgments, and data-efficient learning (due to smaller training set). However, while this claim maybe true, as it stands, in my opinion, it may not be true or inconclusive (please see weaknesses).

---

> ### Author Response · Authors · 2022-08-21
> **Response to Reviewer XFZ1**
>
> **Q1:** Different wording choices in causality.
>
> **A1:** Thanks for the suggestion. We have revised the paper to address the unclarity. As for "how humans interpret A is responsible for B differently from A causes B," such statements can be different in subtle ways when there are multiple "causes" for an event. One intuitive example is "The fall of Lehman brothers is responsible for the financial crisis." vs. "The fall of Lehman brothers is the cause of the financial crisis." In the physical domain, let's consider that ball A is moving towards ball B, but it collides with ball C which slightly changes its trajectory but ball A still collides with B. In such cases, C may be considered responsible but probably not the cause for A-B collision. Throughout the paper and our labeling process, we have kept the wording "be responsible for" to be consistent with the original CLEVRER dataset.
>
> **Q2:** Use language to evaluate machine understanding.
>
> **A2:** Thanks for the great suggestion. We agree and have updated the corresponding paragraphs in the submission. Using language as a lens has two important advantages. First, as you pointed out, it enables generating "explanations." The CLEVRER dataset was designed for this purpose. It contains questions such as "which of the following events is responsible for ..." At the current point, since we do not have good evaluation metrics for machine-generated physical event descriptions, we choose to keep these explanation questions multiple-choice-based. Second, language is useful for specifying events: A spins; A moves; A moves while spinning. These subtleties can be hard to specify with representations such as bounding boxes in videos. Meanwhile, language descriptions are more friendly to human annotators.
>
> **Q3:** Scale of the dataset.
>
> **A3:** We would like to clarify that CLEVRER-Humans is not a dataset designed for training models from scratch. Instead, it is primarily designed for few-shot finetuning and zero-shot evaluation. If we compare our dataset with a similar human annotated visual question answering dataset CLEVR-Humans, CLEVR-Humans contains 25019 labeled questions (training and validation set, https://cs.stanford.edu/people/jcjohns/iep/), while our CLEVRER-Humans contains 21,167 labels for event pairs. Furthermore, our results have already demonstrated the challenges highlighted by CLEVRER-Humans including the grounding of physical events in free-form language descriptions and the modeling of human causal judgments.
>
> **Q4:** Information on the profile of MTurks.
>
> **A4:** Thanks for pointing this out. We hire the MTurkers with the approved HITs of 1000 or higher. We expect the MTurkers to be the general public who are familiar with the basic crowdsourcing process. When collecting data, we release the tasks in batches, where each HIT contains 30 QA pairs mostly coming from one or two videos. We perform quality check to ensure annotators have sufficient knowledge of English language. We also answer their questions about the annotation process by email. It is expected that most speakers use English as their native language. We have included the details in the supplementary (Appendix D)
>
> **Q5:** Was there any quality checks on human annotations? How many annotators annotated each individual sample? Was any sort of agreement among annotators measured in order to ensure quality checks?
>
> **A5:** Quality checks over CEG node correctness are performed by majority voting. Since we have split the annotation of each video to 3 annotators, and they will see overlapping events and annotate their correctness.
>
> Checks for edge correctness are performed by including additional "quality checking" questions. Specifically, each annotator will see 3 videos and 10 questions for each video. 1 of the video will be from a small and manually-curated dataset by authors, containing 30 videos. The entire answer set will be accepted if and only if the annotators answer those quality-checking questions correctly (more specifically, have a small divergence with our answer).
>
> We have added these details in the dataset annotation sections (Appendix D).
>
> **Q6:** Related work on conditional text generation.
>
> **A6:** Thanks! We have added an additional subsection on conditional text generation.

---

> > ### Author Response · Authors · 2022-08-26
> > **Looking Forward to Your Feedback**
> >
> > Dear Reviewer,
> >
> > Thank you again for the constructive reviews, which have helped us greatly improve the quality and clarity of our paper. In our revision, we have updated the motivation of our work, related work, MTurk profile, and quality check details. We hope our response and new results have been able to address your concerns. As we approach the end of the discussion period, please don’t hesitate to let us know if you have any additional questions or comments!
> >
> > Thanks for your time,
> > Authors

---

> > ### Author Response · Authors · 2022-08-29
> > **Looking Forward to Your Feedback**
> >
> > Dear Reviewer,
> >
> > Thank you again for your time and consideration. We hope our response and new results have been able to address your concerns. As we are approaching the last day of the discussion period, please don’t hesitate to let us know if you have any additional questions or comments!
> >
> > Authors

---

### Official Review · Reviewer_NWDY · 2022-07-28
**CLEVRER Explanatory Task with Human Annotations**

**Rating:** 4
**Confidence:** 5
**Clarity:** The paper is clearly written and very…

**Strengths:**

- This paper introduces the first human-annotated video question answering dataset about physical reasoning and causal relationships.
- The paper is well-structured, clearly written and very easy to follow.
- The paper includes detailed statistical analysis of the generated dataset.

**Weaknesses:**

- They claim that the existing benchmarks lack diversity in terms of events / descriptions, but they still use the same visual data with the CLEVRER.
- One could use synonyms (e.g. collide vs. hit) or different templates (e.g. passive vs. active voice) to increase the diversity of language-component. When I check Figure 6 (f), I see that these event types are actually very similar to each other (e.g. bump, collide and hit). Different verb forms are categorized as different individual categories (e.g. pushes vs. pushed, they should be one single category).
- Could the authors explain the second issue that is mentinoned in the abstract more? They only address this issue on the lines 29-33, but it is not clear "_how_ humans interpret _A is responsible for B_ differently from _A causes B_". This needs more explanation.
- The dataset is small-scale (the conclusion section refers it as a large-scale dataset). I'd rather design this dataset as a zero-shot evaluation platform (pretrain on CLEVRER, evaluate on the generated benchmark, and actually the authors performed this experiment).
- The paper puts emphasis on the causal event graphs (CEGs). The CRAFT benchmark also includes a similar representation which is named as causal graphs. There are no discussions about how they relate to / differ from each other. The paper does not even mention the CRAFT's causal graphs.

**Additional Feedback:**

I think it's a well-written paper, the idea is also good but the paper needs revision. Please see my comments on the weaknesses section.

**Correctness:**

- Please just see the weaknesses section for the claims.
- The dataset is constructed in a sound way, I don't see any problems.
- I think the experiments are well designed, it's good to have pretrain, from scratch, and pretrain-then-finetune experiments.
- Since the data is small-scale and the dataset expanded the original vocabulary, I think it'd be good to experiment with a stronger text encoder like BERT or RoBERTa instead of LSTM.

**Documentation:**

I can't see any resources (e.g. a GitHub repository) for (i) the model implementations and (ii) event description generation process. I think a GitHub repository is a MUST.

**Ethics:**

As far as I've seen, there are no ethical concerns for this benchmark.

**Relation To Prior Work:**

- It could also include a subsection for conditional text generation.
- The paper needs to make a comparison of CEGs and CRAFT's causal graphs.
- CATER should not include any form of questions, please double check.

**Summary And Contributions:**

This work proposes a new physical causal reasoning benchmark similar to the explanatory task of the CLEVRER. They discuss that template-based questions / descriptions lack diversity and causal relationships based on rule-based heuristics differ from human judgments. Different from the original explanatory task, they _iteratively_ collect _human-annotated_ causal event descriptions for a subset of CLEVRER examples. They use this collected data to train a trajectory-based event description generation model. Finally, they apply a refining / filtering process to the generated descriptions with the help of human annotators.

---

> ### Author Response · Authors · 2022-08-21
> **Response to Reviewer NWDY**
>
> **Q1:** The benchmark uses the same video set as CLEVRER.
>
> **A1:** Our main purpose is to increase language diversity and incorporate human causality judgments. Thus, we have decided to keep the videos simple.
>
> **Q2:** Use synonyms and more templates.
>
> **A2:** We would like to clarify that due to the limited space, Figure 3(f) only contains the list of words that are most frequent. Per request, we have merged verbs that only differ in forms by lemmatization. There is an updated (full) list of verbs:
>
> ```
> come, move, change, stop, throw,
> slow, go, travel, begin, spin,
> roll, stand, halt, roll, lose,
> leave, head, want, hurl, enter,
> hit, collide, push, bump, push,
> tag, sideswipe, bounce, strike,
> touch, cause
> ```
>
> We also would like to point out that for some verbs, if they seem to be synonyms (e.g., bump and sideswipe), they can have subtle differences in physical grounding. For example, A bumps into B usually implies that A is moving faster than B and its collision changed the state of B. Furthermore, different tenses of the same verb have different meanings in sentences: "the event that ball A moved is responsible for the collision" is different from "the event that ball A is moving is responsible for the collision." In the former case, ball A does not have to be moving while the collision happens.
>
> Finally, it is possible to hand-craft a lot of rules to handle each individual cases (e.g., bump, sideswipe, roll), but that will require additional hyperparameters for thresholding, and may be hard to align with human perception.
>
> **Q3:** Alignment between heuristics and human judgments.
> **A3:** Please kindly refer to our general response. We have included additional human experiments to illustrate the divergence between different automatic generation protocols and human judgments.
>
> **Q4:** Scale of the dataset.
>
> **A4:** We agree with the reviewer that this dataset is primarily for zero-shot evaluation or few-shot training. We have revised the conclusion section to include this note.
>
> **Q5:** Causal event graphs.
>
> **A5:** Thanks for the pointers. We agree that similar graph structures have been studied in related works, and we have added references in the paper. However, our way of obtaining CEGs, including the cloze tasks and the dense edge labeling protocols are new in this field.
>
> **Q6:** Stronger text encoders.
>
> **A6:** Thanks for the suggestion. We have added an additional CNN+BERT baseline, where we have replaced the LSTM encoder with a pretrained (and fixed) BERT encoder. The results have been added to Table 4. We do not see significant improvements over the CNN+LSTM method, possibly because language pretraining does not help the recognition of new event types. Note that our strongest method ALOE is not compatible with BERT embeddings: it uses a multimodal transformer that takes object embeddings and word embeddings as inputs, and does not have a separate language encoder. Similarly, neuro-symbolic approaches NS-DR and VRDP leverage pretrained semantic parsers and do not jointly learn language encoders.
>
> **Q7:** Code release.
>
> **A7:** We have released the code for baseline models and event description generator at https://github.com/xyang23/CLEVRER-Humans1.0
>
> **Q8:** Related work on conditional text generation.
>
> **A8:** Thanks! We have added an additional subsection on conditional text generation.

---

> > ### Author Response · Authors · 2022-08-26
> > **Looking Forward to Your Feedback**
> >
> > Dear Reviewer,
> >
> > Thank you again for the constructive reviews, which have helped us greatly improve the quality and clarity of our paper. In our revision, we have updated our analysis on verb diversity. We also added an additional human study on the difference between heuristics and human judgments. The manuscript has also been updated based on your suggestions about scale and CEGs, and the code for baselines has been released at https://github.com/xyang23/CLEVRER-Humans1.0. We hope our response and new results have been able to address your concerns. As we approach the end of the discussion period, please don’t hesitate to let us know if you have any additional questions or comments!
> >
> > Thanks for your time,
> > Authors

---

> > ### Author Response · Authors · 2022-08-29
> > **Looking Forward to Your Feedback**
> >
> > Dear Reviewer,
> >
> > Thank you again for your time and consideration. We hope our response and new results have been able to address your concerns. As we are approaching the last day of the discussion period, please don’t hesitate to let us know if you have any additional questions or comments!
> >
> > Authors

---

### Official Review · Reviewer_yCNm · 2022-07-28
**Interesting, useful dataset. Paper might benefit from analysis of failure of popular models.**

**Rating:** 8
**Confidence:** 4

**Strengths:**

- Well written paper
- Well constructed dataset
- Evaluation indicates the dataset is challenging to existing models

**Weaknesses:**

- The paper might benefit from some additional analysis of the gap between current models and what the benchmark offers. One way to do this might be to present illustrative examples of common errors that current models make on the new benchmark.
- I think the performance of humans on the final QA formulation of the data should be included.
- The argument about lower training data size is unclear to me. What would be the performance of models on the CLEVERER dataset if their training set were similarly restricted?

**Additional Feedback:**

-

**Clarity:**

Yes. the paper is fairly easy to understand.
Minor point: In Figure 6, several plots of "Edge score" as the label for the x-axis. I don't believe those are correct.

**Correctness:**

Yes. The data construction method is presented in detail. Along with relevant evaluations.

One evaluation that is missing is the human performance on the finally constructed dataset. While the dataset is constructed using human annotations, there is some final automatic conversion of graphs to questions and threshold based filtering (lines 197 to 200). How can one verify that these choices are appropriate?

**Documentation:**

The provided link provides information about construction of dataset and license.

I did not find a maintenance plan explicitly provided. I am assuming this dataset will be hosted at the provided link forever.

I did not find an explicit intended use description. But I am assuming this dataset is mostly going to be used for benchmarking only.



**Ethics:**

None that I can think of.

**Relation To Prior Work:**

Yes.

**Summary And Contributions:**

The paper presents an extension to the CLEVER dataset augmenting it with causal judgments of physical events. The causal judgments are collected in the form of graphs and then converted to a QA format to test existing methods.  Evaluation shows that existing methods that perform well on CLEVERER, seemingly fail on the proposed dataset. The paper suggests that this is because of improved diversity, human annotated causal judgments and smaller training set not allows models to overfit (I am not too convinced by the third argument. Question below.).

---

> ### Author Response · Authors · 2022-08-21
> **Response to Reviewer yCNm**
>
>
> **Q1:** Analysis of the current model.
>
> **A1:** Thank you for the suggestion. We have added additional error analysis for models in the supplementary (Appendix B.3), including examples. Different models have different failure modes. For NS-DR and VRDP, the program parser cannot effectively generate programs for the CLEVRER-Humans due to the increase in vocabulary. Furthermore, for end-to-end learning models such as ALOE, we have not seen significant improvement brought by the pretraining phase. We believe this is primarily because of the domain gap between human judgment and heuristics-based labeling. Specifically, our human experiments have shown that p(Human | CLEVRER-Heuristic) = 0.62. That is, only 62\% of the event pairs that have been labeled as causal in CLEVRER, are labeled as causal by human annotators. Future work may consider other ways of pretraining, such as pretraining on event recognition, which may be more transferable, and pretraining with other types of heuristics.
>
> **Q2:** Human performance.
>
> **A2:** We agree and have included human performance in the paper (Table 4). Specifically, human achieves 84.5% per-option accuracy on our generated CLEVRER-Humans dataset. The best model studied in the paper (ALOE, with pretraining and finetuning) only achieves 52.7% per-option accuracy.
>
> **Q3:** Training data size.
>
> **A3:** Thanks for the suggestion! We would like to clarify that CLEVRER-Humans is primarily designed for finetuning and testing purposes, other than training from scratch. Meanwhile, we agree with your suggestion and have conducted an additional experiment: training and testing an ALOE model (the best-performing model in Table 3) on a subset of the original CLEVRER dataset, whose size is the same as the CLEVRER-Humans dataset. The model achieves an accuracy of 58.1% on the CLEVRER test set. Note that the same ALOE model trained on CLEVRER-Humans from scratch achieves an accuracy of 51.8% on the CLEVRER-Humans test set. These results suggest that future models should primarily consider using CLEVRER-Humans for few-shot finetuning or evaluation only.
>
> **Q4:** Automatic conversion.
>
> **A4:** The conversion from graphs to questions is simply by randomly sampling positive and negative edges in the CEG. We balanced the proportion of positive and negative pairs so that the derived dataset has the same distribution as CLEVRER.
>
> The threshold-based filtering is based on the description of scores: Choice 4: quite responsible, and Choice 5: extremely responsible.
>
> **Q5:** Maintainance and Intended Use descriptions.
>
> **A5:** Thanks for the suggestion. We have updated the paper to include both sections. The dataset will be hosted and maintained by authors, and it will be primarily used for evaluating models for video question answering and causality understanding in videos.

---

> > ### Author Response · Authors · 2022-08-26
> > **Looking Forward to Your Feedback**
> >
> > Dear Reviewer,
> >
> > Thank you again for the constructive reviews, which have helped us greatly improve the quality and clarity of our paper. In our revision, we have included additional analyses of human performance and current model performance. We have also performed the ablation study on data size. We hope our response and new results have been able to address your concerns. As we approach the end of the discussion period, please don’t hesitate to let us know if you have any additional questions or comments!
> >
> > Thanks for your time,
> > Authors

---

### Author Response · Authors · 2022-08-21
**General Response**


We thank all reviewers for their helpful comments and constructive suggestions. In our general response, we wish to address the comparison between human causal judgment and heuristics-generated labels, which were raised by multiple reviewers.

**Q1:** The issue "causal relationships based on manually-defined heuristics are different from human judgments."

**A1:** Thanks for the suggestion! We have added an additional human study to illustrate the point. Specifically, we compare human answers to the original CLEVRER questions with two commonly-used heuristics for automatic generation of the answer. The method Heuristic uses the original CLEVRER heuristic, where event A is directly responsible for event B if A happens before B and they share a common object. The method Counterfactual-Intervention uses counterfactual inference: we removed objects that appear in A but not in B so that A will not happen in the video, and re-run the simulation. We say event A is responsible for B if event B does not happen without A. The results are visualized in the following three-by-three table. The entry p(X|Y) denotes the fraction of event relations that are annotated as causal by protocol X given that the relations are annotated as causal by protocol Y.

| p(X\|Y)          | Y=CLEVRER | Y=Counterfactual | Y=Human |
|------------------|-----------|------------------|---------|
| X=CLEVRER        | 1.00      | 0.74             | 0.96    |
| X=Counterfactual | 0.89      | 1.00             | 0.54    |
| X=Human          | 0.62      | 0.61             | 1.00    |

Our experiments reveal that all three sets of answers diverge from each other to some degree. In general, the original CLEVRER-Heuristic agrees with human judgments more. Qualitatively, we found that heuristic-generated labels primarily differ from human judgment when the cause is indirect (e.g., the question asks whether the presence of an object is responsible for the collision, rather than the movement of that object), or when the event is not the only cause of another event (e.g., ball A and B will still collide even without the collision between B and C).

---

### Meta-Review · Area_Chair_rrvw · 2022-09-11

**Recommendation:** Accept
**Confidence:** 4

**Metareview:**

In this paper, the authors present video reasoning benchmarks. The feedback from reviewers is largely positive, while concerns mainly focus on the diversity of proposed benchmarks and clarification in the presentation, which the authors have tried to fix during the rebuttal period. I would recommend the authors to carefully address these in the revised version (especially those for NWDY and XFZ1)

Overall, I would recommend this paper to be accepted.

---

### Decision · Program_Chairs · 2022-09-16

Accept